# Impacts of COVID-19 Lockdown on Use and Perception of Urban Green Spaces and Demographic Group Differences

**Shiqi Wang** * and **Ang Li**

School of Architecture and Design, China University of Mining and Technology, Xuzhou 221116, China
* Correspondence: wangshiqi@cumt.edu.cn

**Abstract:** The COVID-19 pandemic triggered unprecedented travel restrictions around the world and significantly altered people's daily behaviors. Although previous works have explored the changes in usage and perceptions of urban green spaces (UGS) before and through the pandemic lockdown, there are certain differences in conclusions for various regions, and demographic group differences are not figured out. Our study aimed to evaluate the impacts of the COVID-19 lockdown on the use and perception of urban green spaces in Xuzhou, China and identify the differences across groups through an online survey of 376 respondents. The descriptive statistical results showed that approximately half reduced UGS visits, and one third reported increased importance of UGS's health benefits, especially in mentality. During the lockdown, the city park and community park were the most common destinations and the well-maintained lawn was regarded as the most valued characteristic, followed by sports facilities and seating facilities. Walking was the most frequent means of travel, while public transport was the least common choice. The regression analysis suggested that age, marriage, living pattern and income have significant influences on usage and perception of UGS. The young and the unmarried were more likely to perceive increased social benefits by visiting UGS compared to before the pandemic. People living alone visited the private garden more frequently, and people from three-generation-families preferred green life streets. Richer people unusually spend more time in UGS, benefited more and had more potential to renew green activities. In addition, more perceived risks related to COVID-19 resulted in higher self-reported health benefits. Finally, the suggestions for encouraging UGS visits during the pandemic lockdown are discussed.

**Keywords:** COVID-19; Urban green space; demographic variable; usage and perception; health benefit



## 1. Introduction

The COVID-19 pandemic caused not only a health crisis but also travel restrictions around the world, especially in China. During the lockdown, people had limited access to urban public places. While considering the positive health benefits of green experiences, it is important and necessary to visit urban green spaces during the lockdown periods. Many studies were conducted to explore how people use and perceive the UGS in order to improve its availability and health benefits. COVID-19 has changed residents' daily activities and health needs, which provides a new background for studies on UGS. Understanding how people visit and perceive the UGS during the pandemic lockdown will provide valuable information for green planning and design aiming to promote health and mental restoration.

### 1.1. COVID-19 and Mental Health

The COVID-19 pandemic outbreak in March 2020 was regarded as an international health emergency by the World Health Organization [1] (WHO, 2020). Various policy measures were taken to stop the virus, such as stay-at-home orders, closing schools and workplaces and limited access to public places [2]. These restrictions seriously impeded

people's social interaction, leisure activities and business activities, with negative consequences for their mental health and subjective wellbeing [3]. During the COVID-19 lockdown, people suffered higher mental health risks, such as depression, anxiety and decreased cognitive ability [4] This situation was exacerbated in areas with denser housing, tighter restricted policies, higher infective risk or less accessible services [5].

### 1.2. Health Benefits and Urban Green Spaces

The health benefits derived from urban green spaces (UGS) are well established in the existing research. The UGS refer to one patch of land covered by vegetation in a city with various sizes, plant types, facilities and services, such as parks, forests, gardens and green paths [6]. The exposure to UGS has positive effects on reduced risk of all-cause and circulatory disease (Mitchell and Popham, 2008), less postoperative recovery time [7], lower levels of obesity ([8], increased longevity [9] and less mental disturbances or illness [10]. These health promoting functions may be attributed to physical and mental restoration aroused by natural experiences, such as stress relief and negative emotion modification [11], the support of social interaction [12], more opportunities for physical activity [13] and the supply of ecosystem services such as purified air [14].

The COVID-19 crisis demonstrates an urgent need for UGS [15]. People experience higher levels of stress caused by social isolation, potential health issues and limited outdoor activities during the pandemic lockdowns [16], and therefore, the physiological and psychological benefits of UGS show more significance. A nationwide survey study in Italian shows that the exposure to UGS or other kinds of green features has significant associations with a lower increase in anxiety, fear, sleep disturbance and other negative emotions or mental problem happening during the COVID-19 lockdown [17]. A private garden will promote life satisfaction and subjective well-being in times of COVID-19 [18]. The government encouraged people to spend more time in outdoor green spaces, while complying with the travel restriction policies, in some countries such as Belgium [19]. Moreover, there is evidence that park use has an association with decreased residual case rates, and park visits are regarded safer than other kinds of mobility [20]. Consequently, the UGS plays an important and irreplaceable role on public health and social well-being during the health crisis, as it minimizes inflective risk and offers a restorative experience.

### 1.3. Use Behaviors, Perception and UGS

The use and perception of UGS depend on multiple factors in published studies during non-pandemic periods [21,22]. The use behavior usually mentions usage frequency, time spent and activity intensity [23,24]. Meanwhile a positive perception will enhance patronage and a negative one can prevent visitation and change use patterns [25]. The environmental characteristics play an important role in people's decision to visiting a UGS, including size [26], shape [27], plant species [28], vegetation cover [29], facilities [30] and so on. The distance to green space is also regarded as an important factor [31]. Many studies prove that the visiting decision and perceived results are also related to demographic characteristics such as gender [32,33], age [34], family [35], education level [36], income [37] and cultural background [38].

Furthermore, the pandemic changed the use and perception of UGS [39]. A large body of studies emerged recently to discuss the new changes in people's green behaviors [40]. Most research shows an increased visiting frequency after the pandemic [41]. In addition, some UGSs have new users or re-engaged users [42]. The forest recreation in Bonn (Germany) has a new set of visitors, including youth, families with children and non-locals during the pandemic. The change in green behavior differs in various regions, such as more people choose to walk to small gardens nearby in Italy, while more people visit greenspace in the city suburbs in Lithuania [43]. Additionally, people's opinions on valued environmental features related to restorative perception, travel concern and importance of UGS for their health also changed [44]. There is also evidence of decreased visiting in some regions [45]. The lack of available greenspace, unequal distribution, closed facilities,

various policies, feelings of unwelcome and concerns about infection may prevent people from visiting the UGS [46]. As a result, there is controversy about the change in UGS use caused by the pandemic, which varies in different countries or regions all over the world.

### 1.4. Hypotheses and Aims

Due to an increasing need for mental restoration by visiting UGS during the post-pandemic, we need to understand how the use behaviors and restorative perception in UGS change driven by COVID-19, as well as how the change mechanism varies with demographic factors. Although a body of published research has explored the new use and perception patterns of UGS during the pandemic, detailed discussions on the driving mechanism are limited, especially in China. Most cities in China made strict restrictive policies that the green spaces were closed and people were restricted to their communities or even homes [47] during the early stages of the COVID-19, which may have stimulated stronger restorative demands and more significant behavior changes. When the COVID-19 outbreak was basically under control with no new deaths, the restricted policies continued but were relaxed. Compared to the early stages of the pandemic, this period lasted much longer, and people had limited opportunities to visit public spaces. To better understand the specific issues and demands, the use behaviors and perceptive pattern of UGS in China during the late stages of COVID-19 (April to July 2021) were explored and compared to the cases during the non-pandemic period. We aimed to investigate the changes in use and perception caused by COVID-19. We also analyzed how changes varied with demographic characteristics to explain different social groups' demand and behaviors. The results will be beneficial for guiding the planning and design of UGS to deal with mental restoration after the health crisis.

## 2. Methods

### 2.1. Study Area

Xuzhou was selected as the study area, located in Jiangsu province, eastern China, with a population of approximately 8.8 million people (Figure 1). Under a typical temperate monsoon climate, it has similar plant characteristics to most cities in the mid-China region. By 2019, the coverage of UGS was 43.7% in the built-up area, and the park green space per capita was 15.4% [48]. Xuzhou's GDP ranks in the middle, and its urban green construction is also in step with most cities of similar sizes. Additionally, unlike most cities, a small number of cities were severely affected by the outbreak (such as Wuhan) and were taken out of our consideration. Thus, we think Xuzhou is more representative of most cities in China in terms of its vegetation characteristics, urban development and the effects of the COVID-19 pandemic.

From March to April 2022, Xuzhou went through a period of pandemic lockdown with strict restrictive policies, including closing schools, shops and other public spaces, in addition to outdoor green spaces, and instituting home quarantine orders. The travels of most residents were limited within neighborhood units. Then, as new cases declined, people obtained limited opportunities to visit crucial public spaces with negative results on a 2019-nCoV test. During this period, most cities in China had similar reaction mechanisms and policies to deal with the recurrent outbreaks [49]. Our survey was conducted during the beginning of the restriction policies becoming relaxed but before non-essential businesses were reopened, when people could visit the UGS conditionally and the negative effects of the crisis on health were remarkable.

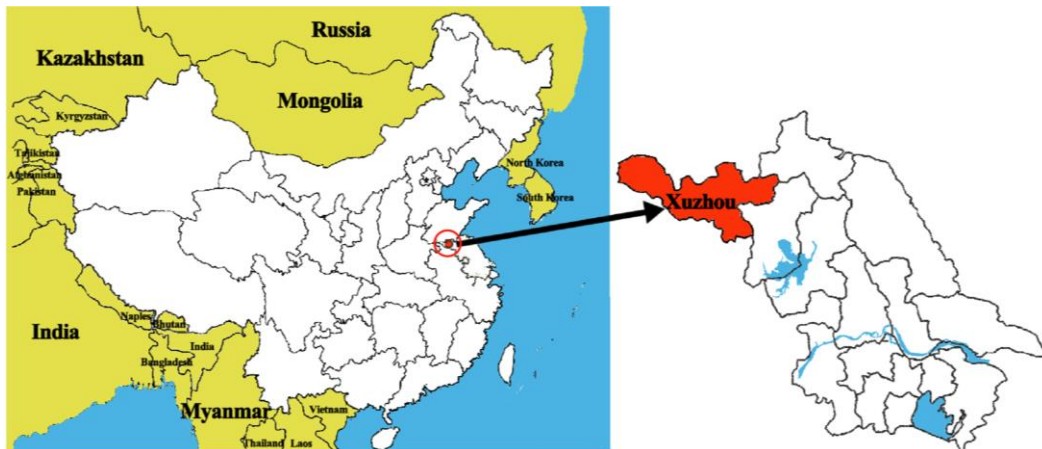

**Figure 1.** The location of Xuzhou.

*2.2. Survey Design*

An online survey was developed and distributed to residents in Xuzhou by using random sampling and the snowball approach. We sent the questionnaire to initial participants through social media (Wechat) and they spread the survey to people around. All participants were required to be over 18 years old and live in the main urban district of Xuzhou city for more than 1 year. People who were experiencing serious illness, life changes or alcohol addiction were excluded to avoid an extremely special response. Participants were notified of the survey objective, data use and their own rights. They were voluntary and allowed to quit the survey at any point.

The questionnaire was made up of three sections. The first part asked about the use and perception of UGS during lockdown, including (1) the type of visited UGS, (2) usage frequency, (3) time spent, (4) main activity, (5) way to travel, (6) perceived importance of UGS's health benefits, (7) self-reported health benefits by visiting UGS, (8) valued environmental characteristics during pandemic visiting and (9) barriers to UGS. The second part investigated the changes before and through the pandemic for the first six items in Section 1. Finally, questions in Section 3 asked about demographic characteristics, including (1) gender, (2) age, (3) yearly household income, (4) educational level, (5) job, (6) marriage, (7) home ownership, (8) residential pattern, (9) income change caused by COVID-19 and (10) perceived health risk of COVID.

These variables, except self-reported health benefits and perceived importance of UGS's health benefits, are categorical. In order to conduct the follow-up quantitative analysis, we coded demographic characteristics (such as for gender, male = 1, female = 2; for age, 18–30 years old = 1, 31–60 years old = 2, over 60 years old = 3). The variables of changes in the frequency, the time spent, the perceived importance of UGS's health benefits consisted of three categories, where 0 = no change, 1 = increased during the lockdown, 2 = decreased during the lockdown. The variables of the type of UGS visited, the main activity taken part in the UGS and the means of travel were coded as "no change" and "change". Furthermore, those two continuous variables were measured by a 5-point Likert Scale.

The pre-research was conducted with 30 samples including people of different ages, genders and educational levels to verify the clarity and legibility of the question statement and the reliability and validity of the scale questions. Modifications were implemented according to suggestions. The questionnaire was proven to have good reliability and validity (Alpha = 0.890, KMO = 0.950).

*2.3. Data Analysis*

The descriptive analyses were conducted to detect how people use and perceive the UGS during lockdown and the difference from before the pandemic. After that, the demographic variables related to the use and perception of UGSs were identified by using

the Chi-square test among different groups. Then, we conducted a serious of regression analyses to further explore the effects of demographic characteristics on the UGS use and perception. The dependent variables concerned the use and perception of UGS, and the independent variables consisted of demographic characteristics. The SPSS 20.0 conducted all analyses.

## 3. Results

### 3.1. Sample Characteristics

The sample consisted of 376 individuals after the exclusion of records with missing information or obviously incorrect answers (*n* = 42). The demographic characteristics are shown in Table 1. The percentage of female participants was a little higher than males, which was consistent with the urban population characteristics of Xuzhou city. The majority were in the age range of 18–60 years old and had completed high education. This could be due to lower access to the online survey for people with low education or advanced ages. More than half of the participants were married or in a couple, had a full-time job and fixed abode. Nearly half of the people had two dwelling patterns, and other forms of living accounted for similar proportions. The incomes for 57% of the sample were negatively affected by the COVID-19 crisis and the overwhelming majority agreed that outbreaks pose varying levels of risk to health.

**Table 1.** Demographic characteristics of samples (*n* = 376).

| Demographic Characteristics | Variables | Number | Percentage (%) | Percentage of Xuzhou Population (%) |
|---|---|---|---|---|
| Gender | Male | 164 | 43.6 | 49.7 |
| | Female | 212 | 56.4 | 50.3 |
| Age | 18~30 years | 170 | 45.2 | 39.7 (below 30 years) |
| | 31~60 years | 180 | 47.9 | 46.5 |
| | Over 60 years | 26 | 6.9 | 13.8 |
| Annual income | Below 50,000 | 64 | 17.02 | - |
| | 50,000–150,000 | 142 | 37.77 | - |
| | 150,000–250,000 | 114 | 30.32 | - |
| | 250,000–350,000 | 24 | 6.38 | - |
| | Over 350,000 | 32 | 8.51 | - |
| Educational level | Secondary education | 92 | 24.47 | 21.2 |
| | Undergraduate | 144 | 38.30 | 43.3 |
| | Postgraduate | 140 | 37.23 | 35.5 |
| Job | Full time job | 242 | 63.8 | - |
| | Part time job | 18 | 4.8 | - |
| | No job | 80 | 21.3 | - |
| | Retirement | 36 | 9.6 | - |
| Marriage | Unmarried | 138 | 36.7 | - |
| | Married or in a couple | 224 | 59.57 | - |
| | Divorced or Widowed | 14 | 3.72 | - |
| Home ownership | Fully owned | 170 | 45.21 | - |
| | Loan to own | 104 | 27.66 | - |
| | Rent at market rate | 54 | 14.36 | - |
| | Subsidized rental | 48 | 12.77 | - |
| Residential pattern | Living alone | 76 | 20.21 | - |
| | Living with contemporary | 82 | 21.81 | - |
| | Two generation dwelling pattern | 168 | 44.68 | - |
| | Three generation dwelling pattern | 50 | 13.3 | - |

**Table 1.** *Cont.*

| Demographic Characteristics | Variables | Number | Percentage (%) | Percentage of Xuzhou Population (%) |
|---|---|---|---|---|
| Income change caused by COVID-19 | Revenue decline | 216 | 57.45 | - |
| | Revenue unchanged | 158 | 42.02 | - |
| | Revenue Increase | 2 | 0.53 | - |
| Perceived health risk of COVID | Not at all | 18 | 4.79 | - |
| | Few | 32 | 8.51 | - |
| | Some | 156 | 41.49 | - |
| | Many | 98 | 26.06 | - |
| | Great many | 72 | 19.15 | - |

### 3.2. Changes before and during the Pandemic

The description of differences between pre-pandemic times and during the lockdown is shown in Figure 2. The majority did not change their main activity played in the UGS and the means of travel (the proportions were 70.2% and 76.6% respectively). Among the group whose behavior changed in both regards, the main change modes were from vigorous physical exercises such as running and ball sports to taking a walk (5.2%), from public transport to walking (11.2%) and from by car to on foot (5.3%). Approximately 40% changed the type of UGS visited frequently. Changes from urban park to community park and changes from urban park to landscape trail were reported most. Those who reduced the frequency and duration of visits were in the majority (56.4% and 42.1%). Approximately one third reported no changes in these two terms. Some people thought the perceived importance of UGS's health benefits on three dimensions increased during the lockdown (35.7% for mental health, 32.5% for physical health and 19.5% for social health). The majority reported no changes happened in this aspect (61.6% for mental health, 62.3% for physical health and 72.7% for social health).

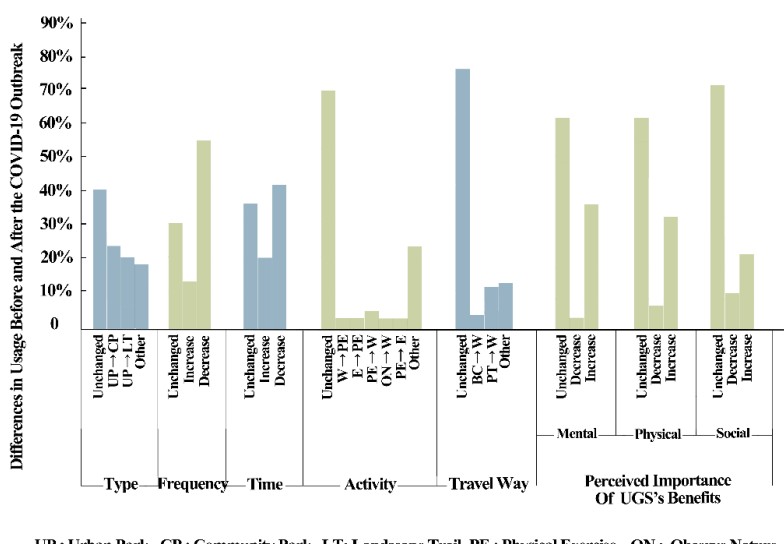

**Figure 2.** Percentage of variation in the type, frequency, duration, activity, means of travel and perceived importance of health benefits during the lockdown relative to before the pandemic.

### 3.3. Use and Perception of UGS during the Lockdown

Figure 3 showed the usage of UGSs during the lockdown. The most common types of UGS visited during the lockdown were city parks (36.8%) and community parks (31.6%). The majority went to the UGSs once or twice a week (37.9%) and stayed 30 to 60 min (52.6%). The most popular activity was taking a walk (46.3%), and the least people chose to participate in recreational or social activities such as dancing, singing and watching shows

(7.4%). Walking (47.4%) was selected as the most common way to travel to the UGSs and public transit (8.4%) had the fewest votes.

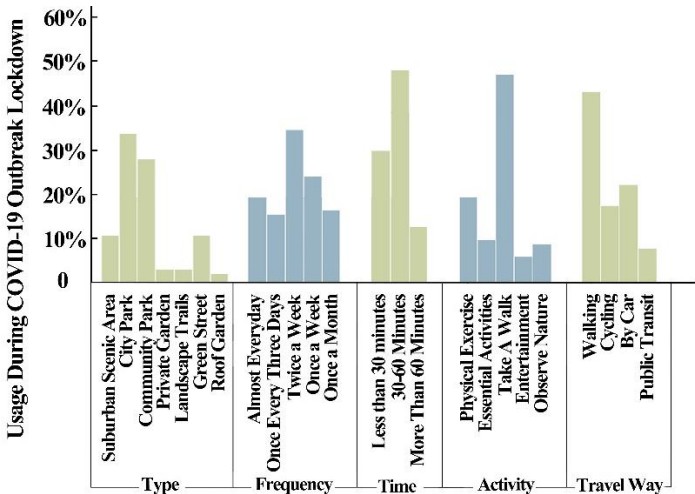

**Figure 3.** Percentage of usage of the UGSs during the lockdown including types of green spaces visited, visit frequency, time spent, main activity and means of travel.

Figure 4 shows the perceptive features of UGSs during the lockdown. More than 80% people thought highly of UGSs when it comes to the mental and physical health; however, fewer people agreed on the value of UGSs for social health (72.3%). Approximately half of the sample reported that visiting the UGSs had brought many or a great many benefits, including enhancing physical exercise (48.0%), improving the mood (60.4%), promoting sociability (31.2%), providing fresh air (64.9%) and promoting family relationships (46.1%). More people thought visiting the UGSs seldom had benefits for socializing (38.7%). The well-maintained lawn was regarded as the most valued characteristic for the green experience during the lockdown by 70% of respondents. Approximately half of the people selected "sports facilities" (49.3%) and "seating facilities" (51.3%). The features written in by about one third of responds were support for socializing (29.9%), places for children (38.3%), good guidance (31.8%), water features (33.1%) and epidemic prevention management (32.5%). The common barriers to visiting the UGSs were "concerns about the COVID-19" (56.4%), "have no enough energy or time" (43.6%) and "lack of accessible green spaces" (41.5%).

### 3.4. Differences in Usage and Perception of UGSs Caused by Demographic Characteristics

For the aspect of the changing situation before and after the pandemic, the results of Chi-square tests showed that there were significant differences in perceived importance of the mental health benefits ($\chi2 = 24.550$, $p = 0.006 < 0.01$) and social health benefits ($\chi2 = 34.488$, $p = 0.000 < 0.01$) obtained by visiting the UGSs among different age groups, in the perceived importance of social health benefits of UGSs ($\chi2 = 13.208$, $p = 0.010 < 0.05$) across the marriage, and in the activity changes ($\chi2 = 8.611$, $p = 0.013 < 0.05$) across income changes. By comparing the percentage (Figure 5a), we found that people between 18 and 30 perceived more importance in UGS for mental health and social health. Unmarried people obtained more social benefits by visiting UGSs (Figure 5b). The majority with an unchanged income did not change their activity types, while people with decreased income were more likely to choose new activity (Figure 5c).

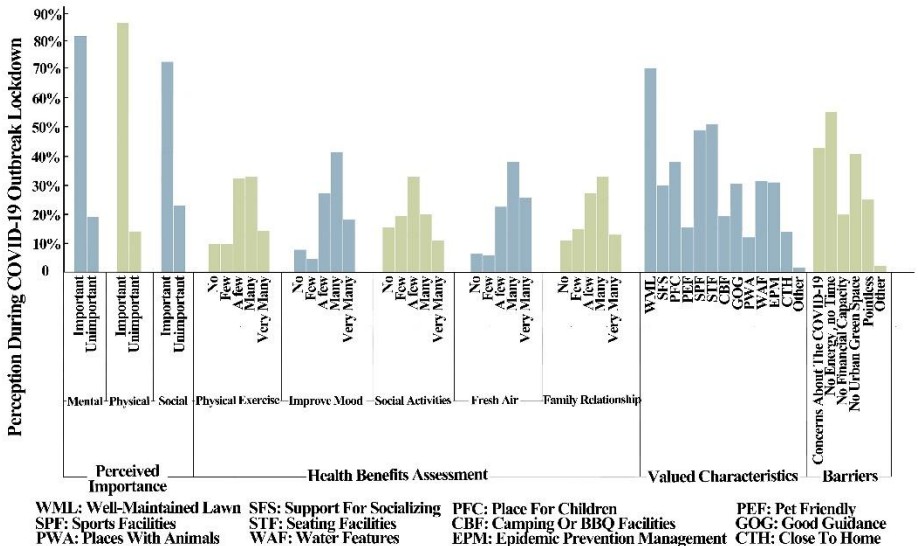

**Figure 4.** Percentage of perception of the UGSs during the lockdown including importance assessment of UGS's health benefits, health benefits obtained by visiting the UGSs, valued characteristics for a green experience during the lockdown and barriers to UGS.

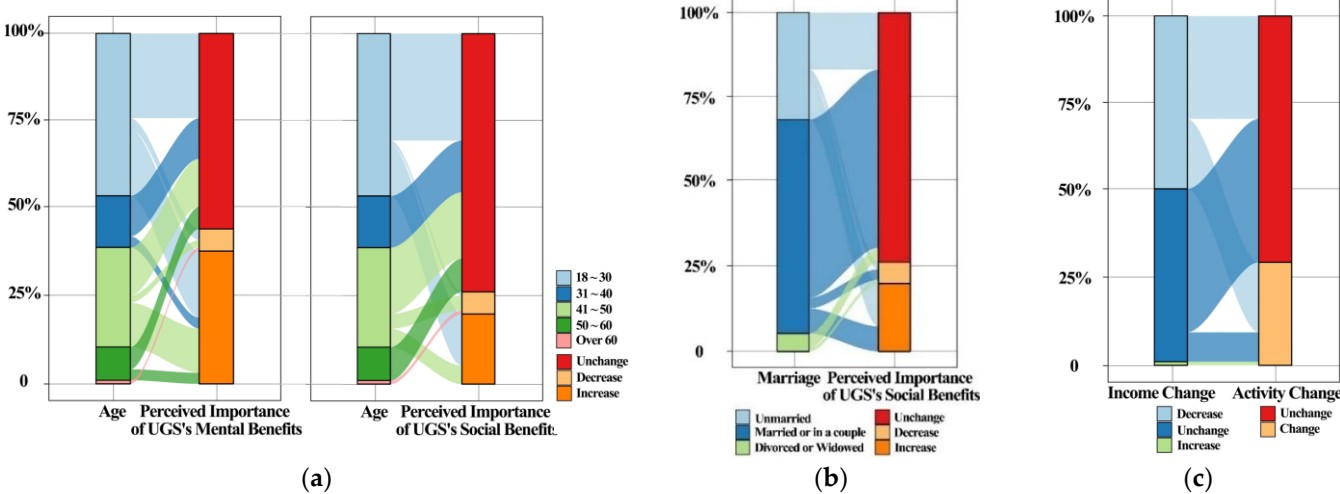

**Figure 5.** Cross figure of certain demographic characteristics and dependent variables related to changes. (**a**) Age and perceived importance of mental and social benefits, (**b**) Marriage and perceived importance of social benefits, (**c**) Income change and activity change.

For the aspect of the usage and perception during the lockdown, the results of Chi-square tests showed that the time spent ($\chi2 = 16.380$, $p = 0.037 < 0.05$) and obtained health benefits in enhancing the physical activities ($\chi2 = 31.938$, $p = 0.010 < 0.05$) varied across annual household income. Lower income groups were more likely to spend less time in the green spaces (Figure 6a). Participants whose annual household income reached over 35 million yuan more frequently regarded that the green experience greatly promoted physical exercises (Figure 6b). There were also some differences in perceived health benefits of enhancing family relationship across educational levels ($\chi2 = 16.897$, $p = 0.031 < 0.05$). The percentage of respondents who selected "many" among the secondary educational level group were obviously lower than the other two groups, and their more common choices were "few" or "a few" (Figure 6c). Furthermore, the various residential models led to different choices in UGS types ($\chi2 = 34.833$, $p = 0.010 < 0.01$). People living in a three-generation-family and living alone were more likely to visit life streets with good greening, and respondents with partners visited community parks or city parks more

frequently (Figure 6d). Additionally, the subjective assessment is that the health risks of COVID-19 caused the different perception in health benefits from visiting the UGSs in the aspects of enhancing physical exercise ($\chi2 = 31.369$, $p = 0.012 < 0.05$), motivating positive moods ($\chi2 = 27.183$, $p = 0.039 < 0.05$) and providing fresh air ($\chi2 = 28.111$, $p = 0.031 < 0.05$). Overall speaking, people who worried significantly about the crisis tended to choose "get much benefits from green experience", and people who considered the risks to be few were more likely to choose "get some benefits from green experience" (Figure 6e).

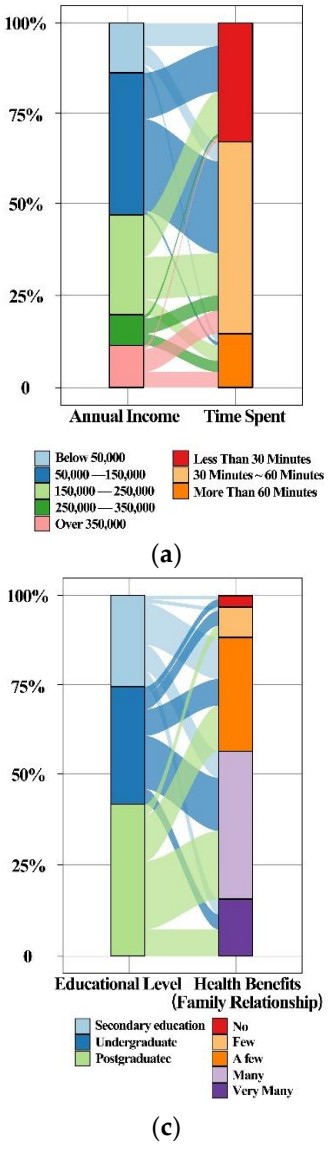

(a)

(c)

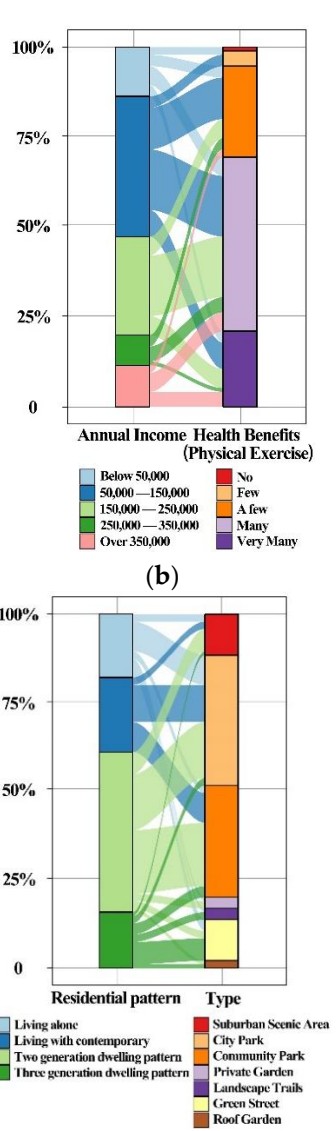

(b)

(d)

**Figure 6.** *Cont.*

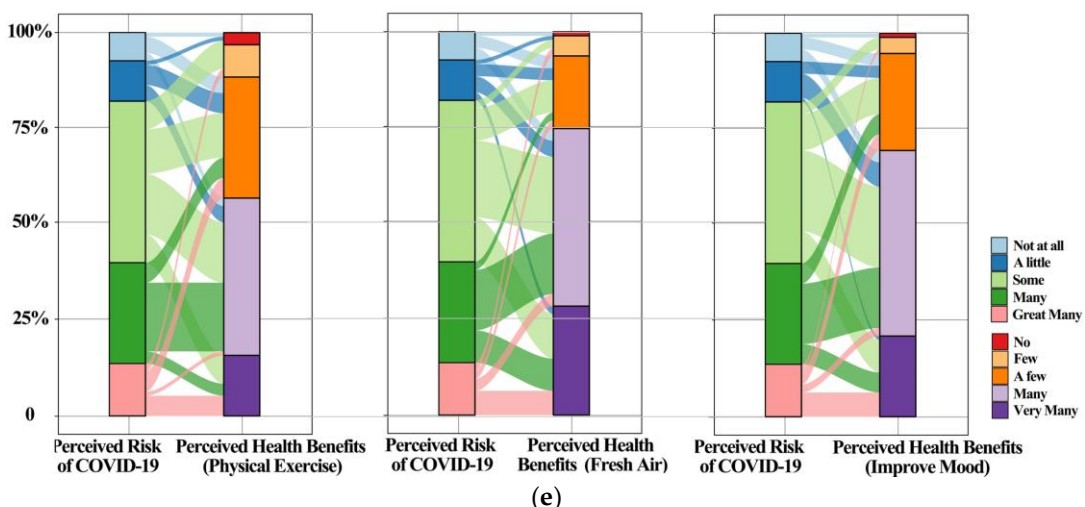

(**e**)

**Figure 6.** Cross figure of demographic characteristics and usage and perception during the lockdown. (**a**) Income and duration, (**b**) Income and health benefits in physical exercise, (**c**) Education and health benefits in family relationship, (**d**) Residential pattern and UGS type, (**e**) Perceived risk of COVID-19 and perceived health benefits.

To sum up, the demographic characteristics, including age, yearly household income, residential pattern, income change and perceived health risk, may have effects on the usage and perception of the UGSs (Figure 7).

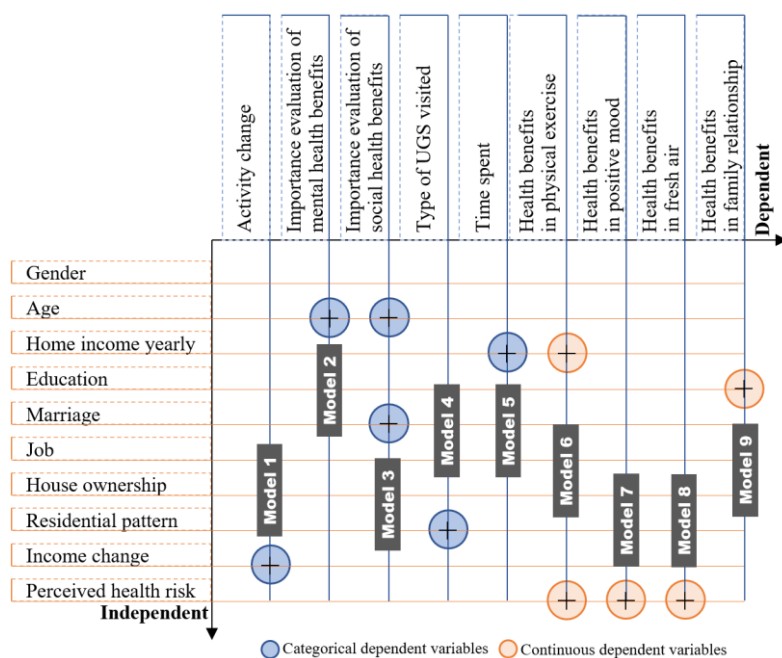

**Figure 7.** Significant associations between demographic characteristics and usage and perception of UGSs according to Chi-square tests (Models 1–9 are built for the following logistic regression analysis.).

### 3.5. Demographic Characteristics and UGS's Use and Perception

Model 1 to Model 5 were conducted using the ordinal regression analysis, as the dependent variables were categorical, and their detailed results are displayed in Tables 2–6. Table 2 demonstrates that age had significant effects on perceived importance of social health benefits. Those aged 18 to 30 were significantly more likely to choose "increased", and the probability of occurrence was approximately 17 times the reference group. Table 3 shows that the yearly household income affects the time spent in the UGSs during the

lockdown, and people from families earning less than 250,000 yuan a year tend to spend less than 60 min in the green spaces. Table 4 states that individuals are more likely to believe the UGSs more important for their social health compared to before if they were unmarried compared to divorced or widowed. Table 5 illustrates that residential pattern had a significant impact on the type of "community park" and "roof garden" relative to the "city park". People living alone tended to visit the private garden compared to people with three generation dwelling patterns. Furthermore, participants from three-generation-families were more likely to visit a green life street in comparison with ones living alone and from two-generation-families. Table 6 explains the obvious effects of income change on the activity change. Individuals were more likely to keep participating in the same activities as before if their income declined compared to increased.

**Table 2.** Change in perceived importance of social health benefits explained by age.

| Variables | Unchanged vs. Decreased | Increased vs. Decreased |
|---|---|---|
| 18–30 years old | 1.000 ($2.248 \times 10^{13}$) | 0.000 ** ($1.472 \times 10^{12}$) |
| 31–40 years old | 1.000 ($1.321 \times 10^{20}$) | 0.994 ($3.330 \times 10^{12}$) |
| 41–50 years old | 1.000 ($3.489 \times 10^{12}$) | 0.981 ($1.315 \times 10^{12}$) |
| 51–60 years old | 1.000 ($1.321 \times 10^{20}$) | 0.995 ($3.330 \times 10^{12}$) |
| Over 60 years old | - | - |

N = 376, $R^2$ = 0.319, Reference system is "decreased social health benefits", ** $p < 0.01$, relative occurrence ratio is in parentheses.

**Table 3.** Time spent in UGSs during the lockdown explained by yearly household income.

| Variables | 30–60 min vs. <30 min | >60 min vs. <30 min |
|---|---|---|
| <50,000 yuan | 0.143 (0.135) | 0.045 * (0.042) |
| 50,000–150,000 yuan | 0.205 (0.195) | 0.004 ** (0.005) |
| 150,000–250,000 yuan | 0.066 (0.085) | 0.016 * (0.024) |
| 250,000–350,000 yuan | 0.809 (0.651) | 0.421 (0.195) |
| >350,000 yuan | - | - |

N = 376, $R^2$ = 0.319, Reference system is "decreased social health benefits", * $p < 0.05$, ** $p < 0.01$, relative occurrence ratio is in parentheses.

**Table 4.** Change in importance evaluation of social health benefits, explained by marriage.

| Variables | Decreased vs. Unchanged | Increased vs. Unchanged |
|---|---|---|
| Unmarried | 0.607 (0.500) | 0.000 ** (116,676,430.2) |
| Married or in a couple | 0.260 (0.240) | 0.240 (21,779,600.3) |
| Divorced or widowed | - | - |

N = 376, $R^2$ = 0.319, Reference system is "decreased social health benefits", ** $p < 0.01$, relative occurrence ratio is in parentheses.

**Table 5.** The type of UGSs visited during the lockdown explained by residential pattern.

| Variables | Suburban Scenic Spot | Community Park | Private Garden | Landscape Trail | Green Life Street | Roof Garden |
|---|---|---|---|---|---|---|
| Living alone | 0.634 (0.500) | 0.138 (0.167) | 0.000 ** (17,941,299.60) | 0.154 (0.125) | 0.050 * (0.125) | 0.997 ($8.852 \times 10^{-9}$) |
| Living with contempo-rary | 0.527 (0.400) | 0.541 (0.533) | 1.000 (0.880) | 0.996 ($6.130 \times 10^{-9}$) | 0.993 ($7.492 \times 10^{-9}$) | 0.997 ($8.173 \times 10^{-9}$) |

**Table 5.** *Cont.*

| Variables | Suburban Scenic Spot | Community Park | Private Garden | Landscape Trail | Green Life Street | Roof Garden |
|---|---|---|---|---|---|---|
| Two generation dwelling pattern | 0.865 (0.800) | 0.775 (0.756) | 0.216 (19,137,386.24) | 0.994 ($9.647 \times 10^{-9}$) | 0.005 ** (0.044) | 0.209 (0.133) |
| Three generation dwelling pattern | - | - | - | - | - | - |

N = 376, $R^2$ = 0.319, Reference system is "decreased social health benefits", * $p < 0.05$, ** $p < 0.01$, relative occurrence ratio is in parentheses.

**Table 6.** Change in activity before and after the pandemic explained by the change of income.

| Variables | Unchanged |
|---|---|
| Decreased income | 0.000 ** (24,922,052.60) |
| Unchanged income | 0.318 (82,443,040.07) |
| Increased income | - |

N = 376, $R^2$ = 0.319, Reference system is "decreased social health benefits", ** $p < 0.01$, relative occurrence ratio is in parentheses.

Models 6–9 were conducted with the optimal scale regression analysis, as the dependent variables were continuous. Model 6 had the acceptable goodness of fit ($R^2$ = 0.8154). The ANOVA table showed the model had statistical significance (Sig = 0.037 < 0.05). The variables of yearly household income ($p = 0.000 < 0.01$, Beta = 0.28) and perceived health risk caused by COVID-19 ($p = 0.000 < 0.01$, Beta = 0.243) both had positive effects on self-reported health benefits in enhancing the physical activities (HBPA) (Table 7). According to the quantization diagram of variables (Figure 8), for people whose family annual income was below 250,000 yuan, the HBPA increased with income. From the section of "250,000 yuan–350,000 yuan" to "over 35,000 yuan", it increased to a greater extent. Whereas from the section of "150,000 yuan–25 yuan" to "250,000 yuan–350,000 yuan", a small decreased occurred. Figure 9 showed that when people felt enough health risks related to COVID-19, the HBPA remained unchanged. The ANOVA test showed the significances of Model 7, Model 8 and Model 9 were 0.233, 0.054 and 0.157, which were both over 0.05. Thus, these three models were not statistically significant and were removed.

**Table 7.** Coefficients of Model 6.

| Variables | Beta | SE | DOF | F | Sig. |
|---|---|---|---|---|---|
| Yearly household income | 0.284 | 0.086 | 4 | 10.812 | 0.000 ** |
| Perceived health risk caused by COVID-19 | 0.243 | 0.090 | 3 | 7.311 | 0.000 ** |

Dependent variable is "the assessment of health benefits in enhancing the physical activities", ** $p < 0.01$.

**Figure 8.** The quantization of the variable of yearly household income.

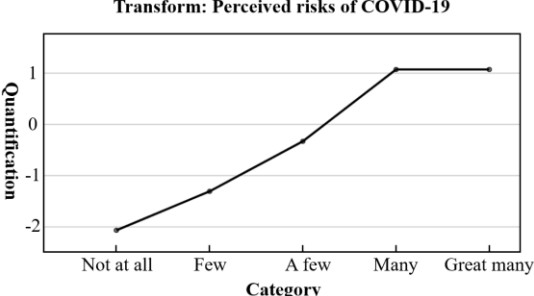

**Figure 9.** The quantization of the variable of perceived health risk caused by COVID-19.

## 4. Discussion

Our survey demonstrated the usage and perception of UGSs during the lockdown and the differences from before the pandemic and the key differences across groups of participants. According to our findings, new evidence was supplied to explain urban residents' use behaviors and perceptive features of the UGSs during the lockdown. In general, the results revealed that the COVID-19 pandemic changed people's behaviors and opinions related to the UGSs, and this change pattern was related to some demographic features.

Our survey showed that some participants considered benefits obtained by visiting UGSs becoming more important for their health across all groups. Notably, more people valued health benefits for psychology more than for physical health and socializing. The results are consistent with the research, which has suggested the greater importance of UGSs in health promotion for people suffering from the health crisis and social isolation[46]. Furthermore, experiencing the UGS was also proven to be more related to mental health relative to physical health and social health [50]. This change pattern of increased social health benefits was particularly obvious among younger participants at the age of 18–30 years old and unmarried people, which supported the indications of increased interest in green spaces among younger people during the lockdown mentioned by Burnett et al., 2022. Older people may suffer from higher risks of social isolation during the lockdown because of the limited ability to use online communications [51].

Some studies reported the increased use of UGS during the pandemic [10], while some showed decreased usage [52]. Most of our survey respondents visited UGSs once or twice a week and spent 30–60 min each time and were more likely to decrease their visit frequency and duration time during the pandemic. The conflicting results may attribute to various survey time, samples, limitation strategies and so on. Furthermore, concerns about COVID-19, lack of time or energy and limited accessibility of UGSs were the most frequently chosen barriers to visiting, which were in line with the published research [53]. Our results indicated that household income was a significant factor for use time of UGS and people with a higher household income tended to spend more time in UGSs during the lockdown. This is supported by previous studies stating that the lower socio-economic position results in less use of green spaces [54].

We also found that although some UGSs, such as community parks and landscape trails, had new visitors during the pandemic, the city park was still the most popular destination. This could be explained by the compensation hypothesis proposed by Maat and De Vries (2006), which holds that people with fewer green spaces nearby tend to visit the further green spaces in compensation [55]. Thus, the eased restrictions that allowed residents more access to urban public spaces contributed to more frequent visits to city parks. The residential patterns were proved to be associated with the choice of UGS. Individuals who lived alone were more likely to visit private parks, and people from three-generation-families were more likely to go to a green street near home. This provides support for those who reported their major companions were family members and children while visiting the UGSs [56]. Additionally, three-generation families consist of older people. Long travel time is usually regarded as the main barrier to visiting the parks by older

people [57]. Movement restrictions enforced due to COVID-19 have stronger negative effects on the elderly. Thus, the streets with good green facilities near home are preferred.

Another key finding is that the main activity carried out in the UGSs changed during the pandemic for the minority but remained unchanged for most participants. This kind of change was obvious among people whose income increased. The published research has reported that higher income contributes to more awareness of benefits from experiencing the UGSs [58]. Thus, people whose income increased had more potential to add investment and proceed with new green activities, such as camping. In addition, taking a walk was regarded as the most common chosen activity in the UGSs, followed by performing physical exercises, according to our results. This is consistent with the numerous research from before the pandemic that pointed out that physical exercise and walking dominated the multiple-activity pattern in UGSs [59]. However, we did not find the statistical significance concerning the change pattern of activity intensity. It may be explained by the proven relationship between user participation and environmental features [60], which were less affected by the pandemic.

Additionally, we found that the yearly household income influenced the perceived health benefits in physical exercise. The overall trend is that perceived health benefits increased as income increased. Although most of the published literature states that those with lower socioeconomic status are more likely to benefit from the UGSs [61], at least the opinions of green gentrification showed that those with high education or high incomes benefited more from neighborhood active green space because they were able to afford pricier properties with better green resources [62]. Moreover, because of travel restrictions, the community green spaces near home have more visitors during the lockdown, which intensifies the differences in health benefits across groups caused by unequal distribution of green resources.

Our results also show the perceived risks associated with self-reported health benefits. Individuals perceiving more health risk from COVID-19 were more likely to obtain physical health benefits from the UGSs during the lockdown. It could be explained by opinions about the need for restoration, which play a moderating role in the mechanism underlying the effects of green experience on health benefits [63,64]. That is, people facing heavier stress both physically and mentally always have stronger restorative needs, and for them, the therapeutic effects of green experiences are more remarkable [65,66]. Notably, the health benefits reported by those who felt COVID-19 was "much risky" and "very much risky" were almost the same. This may illustrate that a higher perceived health risk will no longer trigger an increase in self-reported health benefits, and future research could be conducted to explore the affecting factors in this case.

## 5. Conclusions

The UGSs play an important role in public health, especially for people suffering from stress or crisis. During a lockdown, the government managers and city planners should support urban residents to visit and benefit from the UGS for mental and physical restoration. Considering the risks of infection caused by travel during the lockdown, the UGSs should be attractive enough and compatible with users' needs. Our study proved that the COVID-19 pandemic profoundly affected the usage and perceptive manners of UGSs in Xuzhou, China and identified the differences across groups. The inequalities in accessibility and perceived benefits of UGS were widespread, which might be exacerbated by certain individual factors, such as a miserable marriage, reduced income and lack of companions. Although the research area was limited within Xuzhou, the results could reflect what happened in other cities to a certain extent, which had similar responses to the epidemic. Understanding people's concerns, preference and needs related to UGS during the lockdown could contribute to creating a safe and accessible green experience, as a part of response to COVID-19 or future health crises.

The UGSs should be kept open and available considering their significant effects on health promotion and restoration, especially during the health crisis, on the premise

that the risks they present of spreading the virus are acceptable (Slater et al., 2020). It is potentially difficult to make tradeoffs between encouraging visiting the UGSs and keeping people safe enough. The priorities concerning which spaces should be open or which environmental features should be improved are necessary decisions, for which an understanding of the preferences or demands from urban residents for the UGSs is informative. Our research identified parts of usage patterns and preferences during the lockdown that could supply guidance for the planning, designing or management of UGSs during the pandemic lockdown or a similar public health crisis.

The effects of demographic characteristics on usage and perception of UGSs were well established by much of the previous research [67], while our results emphasized the important roles of age and socioeconomic factors (such as education and income) during the lockdown. Maybe these two factors play a more critical role in where people live or what kind of UGS they have access to. For example, people with a higher socioeconomic status usually enjoy better housing and the elderly are the majority in old communities. When suffering from social isolation, the UGS near home almost becomes the only choice, thus, the needs and expectations of users should be fully considered and taken seriously. The UGSs should be socially inclusive, especially for vulnerable groups, rather than just in the service of the minority. The user-oriented design methods and goals should be developed and adopted by embodying wishes of different groups.

Improving people's awareness of the health-promoting effects of the UGSs will be a major contributor to the use of UGS. Respondents frequently reported not having enough time and experience to travel to UGS in our survey, reflecting an underlying weak desire to visit the UGSs and a lack of understanding and appreciation of the health benefits of UGS. Compared with developed countries, residents in most Chinese cities have lower awareness of health benefits, while the visual–scenic–recreation orientation is highlighted [34]. Residents need to go through a learning process of obtaining clear and consistent messaging about the potential benefits of visiting UGS, in order to build up unbiased green awareness. City planners and managers should carry out various forms of environmental education and publicity measures to communicate the health benefits of UGS to the public effectively.

## 6. Limitations and Future Research

Some limitations exist in our study, which should be overcome in future studies. The method of distributing the questionnaire caused some bias that recruited participants that tended to have certain similar attributes because they came from similar social circles. Some groups of people were excluded from our sample, resulting in the limited generalizability of our results. Cities have experienced different severity and duration of pandemic outbreaks and green spaces in China vary, so indeed the results would change if we chose other cities as the study area. Further research should be conducted to understand how all groups use and perceive the UGS during the lockdown and to explore the differences among cities by collecting more comprehensive samples, carrying on deeper interviews and conducting cross-regional comparative studies. Moreover, the results of our cross-sectional survey could only reflect the characteristics of variables at a single point in time, so a longitudinal study is needed for deeper understanding of the casual relationships between demographic variables and UGS visiting. Additionally, the effects of the environmental features of UGSs on green use and perception were also proven in the published research [68–70], which were not considered and described primarily in our study. The self-reported data concerning the use and perception of UGS lack objectivity. A more comprehensive and accurate influence mechanism should be explored in further research.

**Author Contributions:** Conceptualization, S.W. and A.L.; methodology, S.W.; software, S.W.; validation, S.W.; formal analysis, S.W.; investigation, S.W.; resources, S.W.; data curation, A.L.; writing—original draft preparation, S.W.; writing—review and editing, S.W.; visualization, A.L.; supervision, A.L.; project administration, A.L.; funding acquisition, S.W. All authors have read and agreed to the published version of the manuscript.

**Funding:** The research is supported by Jiangsu Education Department Social Science Projects (2021SJA1023), Doctor of Mass Entrepreneurship and Innovation in Jiangsu Province (JSSCBS20211228).

**Informed Consent Statement:** Informed consent was obtained from all subjects involved in the study.

**Acknowledgments:** We would like to thank the participants in our survey.

**Conflicts of Interest:** The authors declare no conflict of interest.

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
