# Peer review of "Impacts of COVID-19 Lockdown on Use and Perception of Urban Green Spaces and Demographic Group Differences"

_land, doi:10.3390/land11112005_

Round 1
Reviewer 1 Report
The authors have made remarkable contribution in the field. it is accept from my side.Author Response
Thank you for your advice.
Reviewer 2 Report
The manuscript entitled " Impacts of COVID-19 lockdown on use and perception of urban green spaces and demographic groups’ differences" has research value and can enrich such promising approach. I have several issues that have to be addressed before the manuscript can be further evaluated for publication:
- ABSTRACT: The abstract presents a greet and deep description of your work, but the abbreviation UGS needs to be defined in abstract.
- INRTODUCTION:
- The findings of similar studies should be provided.
- Conclusion
- This part presents a greet sections, but the authors should add more results and proposed promising applications.
Reviewer 3 Report
This paper is interesting. I have several comments to improve the quality of this paper.
1. You have a too small sample size. Is there any validity issue? Are they representative?
2. Why did you select Xuzhou? What were the benefits of this selection?
3. In the introduction section, you need to start with why you did this study. Also, what are the contributions that you can make? You focus more on them rather than the literature review.
4. Can you offer a map to describe the study area? I don’t think international readers need to have more ideas on the study area.
5. In this study, you didn’t use weights to survey respondents. Why did you do that? Can you develop weights and use them in your analysis?
6. In the Data Analysis section, you should offer at least one table to describe variables and their descriptive statistics.
7. In Table 1, can you add demographic characteristics of samples as well as the population in the study area to see if your data collection was valid?
8. Did you use the ordinary least square model? If your dependent variable is ordinal, you should employ ordinal regression.
9. To me, the result section looks very unorganized. Can you reformat them?
Round 2
Reviewer 3 Report
Thanks for addressing my comments.